# On Regularization and Robustness of Deep Neural Networks

## Abstract

In this work, we study the connection between regularization and robustness of deep neural networks by viewing them as elements of a reproducing kernel Hilbert space (RKHS) of functions and by regularizing them using the RKHS norm. Even though this norm cannot be computed, we consider various approximations based on upper and lower bounds. These approximations lead to new strategies for regularization, but also to existing ones such as spectral norm penalties or constraints, gradient penalties, or adversarial training. Besides, the kernel framework allows us to obtain margin-based bounds on adversarial generalization. We show that our new algorithms lead to empirical benefits for learning on small datasets and learning adversarially robust models. We also discuss implications of our regularization framework for learning implicit generative models.

## 1 Introduction

Learning predictive models for complex tasks often requires large amounts of annotated data. Interestingly, successful models such as convolutional neural networks are also huge-dimensional and typically involve more parameters than training samples. Such a setting is challenging: besides the fact that training with small datasets is difficult, these models also lack robustness to small adversarial perturbations (Szegedy et al., 2013; Biggio & Roli, 2018). In the context of perceptual tasks such as vision (Szegedy et al., 2013) or speech recognition (Carlini & Wagner, 2018), these perturbed examples are often perceived identically to the original ones by a human, but can lead to arbitrarily different model predictions.

In this paper, we present a unified perspective on regularization and robustness, by viewing convolutional neural networks as elements of a particular reproducing kernel Hilbert space (RKHS) following the work of Bietti & Mairal (2018) on deep convolutional kernels. For such kernels, the RKHS contains indeed deep convolutional networks similar to generic ones—up to smooth approximations of rectified linear units. Such a point of view is interesting to adopt for deep networks since it provides a natural regularization function, the RKHS norm, which allows us to control the variations of the predictive model according to the geometry induced by the kernel. Besides, the norm also acts as a Lipschitz constant, which provides a direct control on the stability to adversarial perturbations.

In contrast to traditional kernel methods, the RKHS norm cannot be explicitly computed in our setup. Yet, these norms admit numerous approximations—lower bounds and upper bounds—which lead to new strategies for regularization based on penalties, constraints, or combinations thereof. Additionally, our framework provides an interpretation of some existing regularization approaches as controlling either upper or lower bounds on the RKHS norm, such as using penalties or constraints on the spectral norm of the filters (Yoshida & Miyato, 2017), various forms of robust optimization (Madry et al., 2018), double backpropagation (Drucker & Le Cun, 1991), and tangent propagation (Simard et al., 1998).

Moreover, regularization and robustness are tightly linked in our kernel framework. In particular, we study the connection between adversarial training under $\ell_2$ perturbations and penalizing with the RKHS norm, in a similar vein to Xu et al. (2009b); from a statistical point of view, we extend margin-based generalization bounds in the spirit of Bartlett et al. (2017); Boucheron et al. (2005) to the setting of *adversarially robust* generalization (see

Schmidt et al., 2018). Following these observations, we provide an empirical evaluation of the different regularization strategies obtained, by considering learning tasks with small datasets or with adversarial perturbations. Importantly, we find that robust optimization approaches often lead to a poor control of the RKHS norm, in favor of local stability and data fit, and that approaches based on spectral norms can be locally unstable. In contrast, our new lower bound penalties, as well as combined approaches based on both upper and lower bounds, enable a tighter control of the RKHS norm, yielding empirical benefits in various regimes and better guarantees. We also provide a new perspective on recent successful approaches to training generative adversarial networks as optimizing a kernel two-sample test (see, *e.g.*, Gretton et al., 2012).

**Related work.** The construction of hierarchical kernels and the study of neural networks in the corresponding RKHS was studied by Mairal (2016); Zhang et al. (2016; 2017); Bietti & Mairal (2018). Our study of the relationship between robustness and regularization follows from Xu et al. (2009a;b), where the main focus is on linear models with quadratic or hinge losses. Some of the regularization strategies we obtain from our kernel perspective are closely related to previous approaches to adversarial robustness (Cisse et al., 2017; Madry et al., 2018; Simon-Gabriel et al., 2018; Roth et al., 2018), to improving generalization (Drucker & Le Cun, 1991; Miyato et al., 2018b; Sedghi et al., 2018; Simard et al., 1998; Yoshida & Miyato, 2017), and stable training of generative adversarial networks (Roth et al., 2017; Gulrajani et al., 2017; Arbel et al., 2018; Miyato et al., 2018a). The notion of adversarial generalization was considered by Schmidt et al. (2018), who provide lower bounds on a specific example of data distribution. Sinha et al. (2018) provide generalization guarantees in the different setting of distributional robustness; compared to our bound, they consider expected loss instead of classification error, and their bounds do not highlight the dependence on the model complexity.

## 2 Regularization strategies for deep neural networks

In this section, we recall the kernel perspective on deep networks introduced by Bietti & Mairal (2018), and present upper and lower bounds on the RKHS norm of a given model, leading to various regularization strategies. For simplicity, we first consider real-valued networks and binary classification, before discussing multi-class extensions.

### 2.1 Relation between deep neural networks and RKHSs

Kernel methods consist of mapping data living in a set $\mathcal{X}$ to a RKHS $\mathcal{H}$ associated to a positive definite kernel $K$ through a mapping function $\Phi : \mathcal{X} \to \mathcal{H}$, and then learning simple machine learning models in $\mathcal{H}$. Specifically, when considering a real-valued regression or binary classification problem, classical kernel methods find a prediction function $f : \mathcal{X} \to \mathbb{R}$ living in the RKHS which can be written in linear form, i.e., such that $f(x) = \langle f, \Phi(x) \rangle_{\mathcal{H}}$ for all $x$ in $\mathcal{X}$. While explicit mapping to a possibly infinite-dimensional space is of course only an abstract mathematical operation, learning $f$ can be done implicitly by computing kernel evaluations and typically by using convex programming (Schölkopf & Smola, 2001).

Moreover, the RKHS norm $\|f\|_{\mathcal{H}}$ acts as a natural regularization function, which controls the variations of model predictions according to the geometry induced by $\Phi$:

$$|f(x) - f(x')| \leq \|f\|_{\mathcal{H}} \cdot \|\Phi(x) - \Phi(x')\|_{\mathcal{H}}. \tag{1}$$

Unfortunately, traditional kernel methods become difficult to use when the datasets are large or when evaluating the kernel is intractable. Here, we propose a different approach that considers explicit parameterized representations of functions contained in the RKHS, given by generic convolutional neural networks, and leverage properties of the RKHS and the kernel mapping in order to regularize when learning the network parameters.

Consider indeed a real-valued deep convolutional network $f : \mathcal{X} \to \mathbb{R}$, where $\mathcal{X}$ is simply $\mathbb{R}^d$, with rectified linear unit (ReLU) activations and no bias units. By constructing an appropriate multi-layer hierarchical kernel, Bietti & Mairal (2018) show that the corresponding RKHS $\mathcal{H}$ contains a convolutional network with the same architecture and parameters as $f$,

but with activations that are smooth approximations of ReLU. Although the model predictions might not be strictly equal, we will abuse notation and denote this approximation with smooth ReLU by $f$ as well, with the hope that the regularization procedures derived from the RKHS model will be effective in practice on the original CNN $f$.

Besides, Bietti & Mairal (2018) show that the mapping $\Phi(\cdot)$ is non-expansive:

$$\|\Phi(x) - \Phi(x')\|_{\mathcal{H}} \leq \|x - x'\|_2, \tag{2}$$

so that controlling $\|f\|_{\mathcal{H}}$ provides some robustness to additive $\ell_2$-perturbations, by (1). Additionally, with appropriate pooling operations, Bietti & Mairal (2018) show that the kernel mapping is also stable to deformations, meaning that the RKHS norm also controls robustness to translations and other transformations including scaling and rotations (Engstrom et al., 2017), which can be seen as deformations when they are small.

In contrast to standard kernel methods, where the RKHS norm is typically available in closed form, this norm is difficult to compute in our setup, and requires approximations. The following sections present upper and lower bounds on $\|f\|_{\mathcal{H}}$, with linear convolutional operations denoted by $W_k$ for $k = 1, \ldots, L$, where $L$ is the number of layers. Defining $\theta := \{W_k : k = 1, \ldots, L\}$, we then leverage these bounds to approximately solve the following penalized or constrained optimization problems on a training set $(x_i, y_i), i = 1, \ldots, n$:

$$\min_{\theta} \frac{1}{n} \sum_{i=1}^{n} \ell(y_i, f_\theta(x_i)) + \lambda \|f_\theta\|_{\mathcal{H}}^2 \quad \text{or} \quad \min_{\theta : \|f_\theta\|_{\mathcal{H}} \leq C} \frac{1}{n} \sum_{i=1}^{n} \ell(y_i, f_\theta(x_i)). \tag{3}$$

We also note that while the construction of Bietti & Mairal (2018) considers VGG-like networks (Simonyan & Zisserman, 2014), the regularization algorithms we obtain in practice can be easily adapted to different architectures such as residual networks (He et al., 2016).

## 2.2 Exploiting Lower bounds of the RKHS norm

In this section, we devise regularization algorithms by leveraging lower bounds on $\|f\|_{\mathcal{H}}$, which are obtained by relying on the following variational characterization of Hilbert norms:

$$\|f\|_{\mathcal{H}} = \sup_{\|u\|_{\mathcal{H}} \leq 1} \langle f, u \rangle_{\mathcal{H}}. \tag{4}$$

At first sight, this definition is not useful since the set $U = \{u \in \mathcal{H} : \|u\|_{\mathcal{H}} \leq 1\}$ may be infinite-dimensional and the inner products $\langle f, u \rangle_{\mathcal{H}}$ cannot be computed in general. Thus, we devise tractable lower bound approximations by considering smaller sets $\bar{U} \subset U$.

**Adversarial perturbation penalty.** The non-expansiveness of $\Phi$ allows us to consider the subset $\bar{U} \subset U$ defined as $\bar{U} = \{\Phi(x + \delta) - \Phi(x) : x \in \mathcal{X}, \|\delta\|_2 \leq 1\}$, leading to the bound

$$\|f\|_{\mathcal{H}} \geq \sup_{x \in \mathcal{X}, \|\delta\|_2 \leq 1} f(x + \delta) - f(x), \tag{5}$$

which is reminiscent of adversarial perturbations. Adding a regularization parameter $\epsilon > 0$ in front of the norm then corresponds to different sizes of perturbations:

$$\epsilon \|f\|_{\mathcal{H}} = \sup_{\|u\|_{\mathcal{H}} \leq \epsilon} \langle f, u \rangle_{\mathcal{H}} \geq \sup_{x \in \mathcal{X}, \|\delta\|_2 \leq \epsilon} f(x + \delta) - f(x). \tag{6}$$

Using this lower bound or its square as a penalty in the objective (3) when training a neural network can then provide a way to regularize. Optimizing over adversarial perturbations has been useful to obtain robust models (*e.g.*, the PGD method of Madry et al., 2018), yet our approach differs in two important ways: (i) it involves a global maximization problem on the input space $\mathcal{X}$, as opposed to only maximizing on perturbations near training data; (ii) it involves a separate penalty term in the training objective, in contrast to PGD which encourages a perfect fit of the training data by considering perturbations on the loss term. We further discuss the links with the robust optimization problem solved by PGD below.

In practice, optimizing over all $x$ in $\mathcal{X}$ is difficult, and we can replace $\mathcal{X}$ with random subsets of examples (such as mini-batches), which may be labeled or not, yielding further

lower bounds on the penalty. In the context of a mini-batch stochastic gradient algorithm, one can obtain a subgradient of this penalty by first finding maximizers $\hat{x}, \hat{\delta}$ (where $\hat{x}$ is one of the examples in the mini-batch), and then simply computing gradients of $f_\theta(\hat{x}+\hat{\delta}) - f_\theta(\hat{x})$ (or its square) w.r.t. $\theta$ using back-propagation. We compute the perturbations $\hat{\delta}$ for each example $x$ by using a few steps of projected subgradient ascent with constant step-lengths, in a similar fashion to Madry et al. (2018) in the context of robust optimization.

**Relationship with robust optimization.** In some contexts, our penalized approach is related to solving the robust optimization problem

$$\min_\theta \frac{1}{n} \sum_{i=1}^n \sup_{\|\delta\|_2 \leq \epsilon} \ell(y_i, f_\theta(x_i + \delta)), \tag{7}$$

which is commonly considered for training adversarially robust classifiers (Kolter & Wong, 2017; Madry et al., 2018; Raghunathan et al., 2018). In particular, Xu et al. (2009b) show that the penalized and robust objectives are equivalent in the case of the hinge loss with linear predictors, when the data is non-separable. They also show the equivalence for kernel methods when considering the (intractable) full perturbation set $U$ around each point in the RKHS $\Phi(x_i)$, that is, predictions $\langle f, \Phi(x_i) + u \rangle_\mathcal{H}$ with $u$ in $U$. Intuitively, when a training example $(x_i, y_i)$ is misclassified, we are in the "linear" part of the hinge loss, so that

$$\sup_{\|u\|_\mathcal{H} \leq \epsilon} \ell(y_i, \langle f, \Phi(x_i) + u \rangle_\mathcal{H}) = \ell(y_i, \langle f, \Phi(x_i) \rangle_\mathcal{H}) + \sup_{\|u\|_\mathcal{H} \leq \epsilon} \langle f, u \rangle_\mathcal{H} = \ell(y_i, f(x_i)) + \epsilon \|f\|_\mathcal{H}.$$

For other losses such as the the logistic loss, a regularization effect is still present even for correctly classified examples, though it may be smaller since the loss has a reduced slope for such points, leading to a more *adaptive* regularization mechanism which may automatically reduce the amount of regularization when the data is easily separable. However, this approach might only encourage local stability, while the global quantity $\|f\|_\mathcal{H}$ may grow uncontrolled in order to better fit the data. Nevertheless, it is easy to show that the robust objective (7) lower bounds the penalized objective with penalty $\epsilon \|f\|_\mathcal{H}$.

**Gradient penalties.** Taking $\bar{U} = \{\frac{\Phi(x)-\Phi(y)}{\|x-y\|_2} : x, y \in \mathcal{X}\}$, which is a subset of $U$ by Eq. (2)—it turns out that this is the same set as above, since $\Phi$ is homogeneous (Bietti & Mairal, 2018) and $\mathcal{X} = \mathbb{R}^d$—we obtain a lower bound based on the Lipschitz constant of $f$:

$$\|f\|_\mathcal{H} \geq \|f\|_L := \sup_{x,y \in \mathcal{X}} \frac{f(x) - f(y)}{\|x - y\|_2} \geq \sup_{x \in \mathcal{X}} \|\nabla f(x)\|_2$$

where the last inequality becomes an equality when $\mathcal{X}$ is convex, and the supremum is taken over points where $f$ is differentiable. Although we are unaware of previous work using such a lower bound for a generic regularization penalty, we note that variants replacing the supremum over $x$ by an expectation over data have been recently used to stabilize the training of generative adversarial networks (Gulrajani et al., 2017; Roth et al., 2017), and we provide insights in Section 3.2 on the benefits of RKHS regularization in such a setting. Related penalties have been considered in the context of robust optimization, for regularization or robustness, noting that a penalty based on the gradient of the loss can give a good approximation of (7) when $\epsilon$ is small (Drucker & Le Cun, 1991; Lyu et al., 2015; Roth et al., 2018; Simon-Gabriel et al., 2018). This has the advantage of overcoming the difficulties of the maximization problem over $\delta$ by leveraging the closed-form expression of the gradient norm, but it may only provide a poor approximation when $\epsilon$ is large, and exhibits the same concerns as other robust optimization approaches in terms of poorly controlling $\|f\|_\mathcal{H}$.

**Penalties based on deformation stability.** We may also obtain new penalties by considering sets $\bar{U} = \{\Phi(\tilde{x}) - \Phi(x) : x \in \mathcal{X}, \tilde{x} \text{ is a small deformation of } x\}$, where the amount of deformation is dictated by the stability bounds of Bietti & Mairal (2018) in order to ensure that $\bar{U} \subset U$. In particular, such bounds depend on the maximum displacement and maximum Jacobian norm of the diffeomorphisms considered. These can be easily computed for various parameterized families of transformations, such as translations, scaling or rotations, leading to simple ways to control the regularization strength through the parameters

of these transformations. One can also consider infinitesimal deformations from such parameterized transformations, and by replacing the supremum over $\mathcal{X}$ by an expectation over training data, we then obtain a gradient penalty that resembles the *tangent propagation* regularization strategy of Simard et al. (1998). If instead we consider robust optimization formulation (7), we obtain a form of *data augmentation* where transformations are optimized instead of sampled, as in Engstrom et al. (2017).

One advantage of these approaches based on lower bounds is that the obtained penalties are independent of the model parameterization, making them flexible enough to use with more complex architectures in practice. In addition, the connection with robust optimization can provide a useful mechanism for adaptive regularization. However, these lower bounds do not guarantee a control on the RKHS norm, and this is particularly true for robust optimization approaches, which may favor small training loss and local stability over global stability through $\|f\|_{\mathcal{H}}$. Nevertheless, our experiments suggest that our new approaches based on separate penalties often do help in controlling upper bounds as well (see Section 4).

## 2.3 Exploiting upper bounds: optimization with spectral norms

In contrast to lower bounds, upper bounds can provide a guaranteed control on the RKHS norm. Bietti & Mairal (2018) show the following upper bound:

$$\|f\|_{\mathcal{H}} \leq \omega(\|W_1\|, \ldots, \|W_L\|), \tag{8}$$

where $\omega$ is increasing in all of its arguments, and $\|W_k\|$ is the spectral norm of the linear operator $W_k$. Here, we simply consider the spectral norm on the filters, given by $\|W\| := \sup_{\|x\|_2 \leq 1} \|Wx\|_2$. Other generalization bounds relying on similar quantities have been proposed for controlling complexity (Bartlett et al., 2017; Neyshabur et al., 2018), suggesting that using them for regularization is relevant even beyond our kernel perspective, as observed in previous work (Cisse et al., 2017; Sedghi et al., 2018; Yoshida & Miyato, 2017).

**Penalizing the spectral norms.** One way to control the upper bound (8) when learning a neural network $f_\theta$ is to consider a regularization penalty based on spectral norms

$$\min_\theta \frac{1}{n} \sum_{i=1}^n \ell(y_i, f_\theta(x_i)) + \lambda \sum_{l=1}^L \|W_l\|^2, \tag{9}$$

where $\lambda$ is a regularization parameter. In the context of a stochastic gradient algorithm, one can obtain (sub)gradients of the penalty by computing singular vectors associated to the highest singular value of each $W_l$. We consider the method of Yoshida & Miyato (2017), which computes such singular vectors approximately using one or two iterations of the power method, as well as a more costly approach using the full SVD.

**Constraining the spectral norms with a continuation approach.** In the constrained setting, we want to optimize:

$$\min_\theta \frac{1}{n} \sum_{i=1}^n \ell(y_i, f_\theta(x_i)) \quad \text{s.t } \|W_l\| \leq \tau \; ; \; l \in 1, \ldots, L \; ,$$

where $\tau$ is a user-defined constraint. This objective may be optimized by projecting each $W_l$ in the spectral norm ball of radius $\tau$ after each gradient step. Such a projection is achieved by truncating the singular values to be smaller than $\tau$ (see Appendix A). We found that the loss was hardly optimized with this approach, and therefore introduce a continuation approach with an exponentially decaying schedule for $\tau$ reaching a constant $\tau_0$ after a few epochs, which we found to be important for good empirical performance.

**Combining with lower bounds.** While these upper bound strategies are useful for limiting model complexity, we found them less effective for robustness in our experiments (see Section 4.2). However, we found that combining with lower bound approaches can overcome this weakness, perhaps due to a better control of local stability. In particular, such combined approaches often provide the best generalization performance in small data scenarios, as well as better guarantees on adversarially robust generalization thanks to a tighter control of the RKHS norm compared to robust optimization alone.

## 2.4 Extension to multiple classes and non-Euclidian geometries

We now discuss how to extend the regularization strategies to multi-valued networks, in order to deal with multiple classes. First, the upper bound strategies of Section 2.3 are easy to extend, by simply considering spectral norms up to the last layer. This is also justified by the generalization bound of Bartlett et al. (2017), which applies to the multi-class setup.

In the context of lower bounds, we can consider a multi-class penalty $\|f_1\|_{\mathcal{H}}^2 + \ldots + \|f_K\|_{\mathcal{H}}^2$ for an $\mathbb{R}^K$-valued function $f = (f_1, f_2, \ldots, f_K)$. One can then consider lower bounds on this penalty by leveraging lower bounds on each individual $f_k$. In particular, we define:

$$\|f\|_M^2 := \sum_{k=1}^{K} \sup_{x \in \mathcal{X}, \|\delta\|_2 \leq \epsilon} (f_k(x + \delta) - f_k(x))^2 \quad \text{and} \quad \|\nabla f\|^2 := \sum_{k=1}^{K} \sup_{x \in \mathcal{X}} \|\nabla f_k(x)\|_2^2,$$

where we use mini-batches of training data instead of $\mathcal{X}$ in our experiments. For robust optimization formulations (7), the extension is straightforward, given that standard multi-class losses such as the cross-entropy loss can be directly optimized in an adversarial training or gradient penalty setup.

Finally, we note that while the kernel approach introduced in this section mainly considers the Euclidian geometry in the input space, it is possible to consider heuristic alternatives for other geometries, such as $\ell_\infty$ perturbations, as discussed in Appendix B.

## 3 Theoretical guarantees and insights

In this section, we study how standard margin-based generalization bounds can be extended to an adversarial setting in order to provide theoretical guarantees on adversarially robust generalization. We then discuss how our kernel approach provides novel interpretations for training generative adversarial networks.

## 3.1 Guarantees on adversarial generalization

While various methods have been introduced to empirically gain robustness to adversarial perturbations, the ability to generalize with such perturbations, also known as *adversarial generalization* (Schmidt et al., 2018), still lacks theoretical understanding and useful guarantees. Margin-based bounds have been useful to explain the generalization behavior of learning algorithms that can fit the training data well, such as kernel methods, boosting and neural networks (Koltchinskii et al., 2002; Boucheron et al., 2005; Bartlett et al., 2017). Here, we show how such arguments can be adapted to obtain guarantees on *adversarial* generalization, *i.e.*, on the expected classification error in the presence of an $\ell_2$-bounded adversary, based on the RKHS norm of a given model learned from data. For a binary classification task with labels in $\mathcal{Y} = \{-1, 1\}$ and data distribution $\mathcal{D}$, we would like to bound the expected adversarial error of a classifier $f$, given for some $\epsilon > 0$ by

$$\text{err}_{\mathcal{D}}(f, \epsilon) := P_{(x,y) \sim \mathcal{D}}(\exists \|\delta\|_2 \leq \epsilon : \; yf(x + \delta) < 0). \tag{10}$$

Leveraging the fact that $f$ is $\|f\|_{\mathcal{H}}$-Lipschitz, we now show how to further bound this quantity using empirical margins, following the usual approach to obtaining margin bounds for kernel methods (*e.g.*, Boucheron et al., 2005). Consider a training dataset $(x_1, y_1), \ldots, (x_n, y_n) \in \mathcal{X} \times \mathcal{Y}$. Define

$$L_n^\gamma(f) := \frac{1}{n} \sum_{i=1}^{n} \mathbb{1}\{y_i f(x_i) < \gamma\}.$$

We then have the following bound, proved in Appendix C:

**Proposition 1.** *With probability $1 - \delta$ over a dataset $\{(x_i, y_i)\}_{i=1,\ldots,n}$, we have, for all choices of $\gamma > 0$ and $f \in \mathcal{H}$,*

$$err_{\mathcal{D}}(f, \epsilon) \leq L_n^{\gamma + 2\epsilon \|f\|_{\mathcal{H}}}(f) + O\left( \frac{\|f\|_{\mathcal{H}}}{\gamma \sqrt{n}} \sqrt{\frac{1}{n} \sum_{i=1}^{n} K(x_i, x_i)} + \sqrt{\frac{\log(C(\|f\|_{\mathcal{H}}, \gamma)/\delta)}{n}} \right), \tag{11}$$

*where $C(\|f\|_{\mathcal{H}}, \gamma) = (1 + 4(\log_2 \|f\|_{\mathcal{H}})^2) \cdot (1 + 4(\log_2(1/\gamma))^2)$.*

When $\epsilon = 0$, this leads to the usual margin bound, while $\epsilon > 0$ yields a bound on adversarial error $\mathrm{err}_{\mathcal{D}}(f, \epsilon)$, for some neural network $f$ learned from training data. We note that other complexity measures based on products of spectral norms may be used instead of $\|f\|_{\mathcal{H}}$, as well as multi-class extensions, following Bartlett et al. (2017); Neyshabur et al. (2018).

One can then study the effectiveness of a regularization algorithm by inspecting cumulative distribution (CDF) plots of the *normalized margins* $\bar{\gamma}_i = y_i f(x_i)/\|f\|_{\mathcal{H}}$, for different strenghts of regularization (an example is given in Figure 2, Section 4.2). According to the bound (11), one can assess expected adversarial error with $\epsilon$-bounded perturbations by looking at the part of the plot to the right of $\bar{\gamma} = 2\epsilon$. In particular, the value of the CDF at such a value of $\bar{\gamma}$ is representative of the bound for large $n$ (since the second term is negligible), while for smaller $n$, the best bound is obtained for a larger value of $\bar{\gamma}$, which also suggests that the right side of the plots is indicative of performance on small datasets.

When the RKHS norm can be well approximated, then our bound provides a certificate on test error in the presence of adversaries. While such an approximation is difficult to obtain in general, the guarantee is most useful when lower and upper bounds are controlled together. We found this to be the case in our experiments for our new lower bound penalties based on adversarial perturbations or gradient norms (see Section 4.2). In contrast, we found that the upper bound is poorly controlled for robust optimization approaches such as PGD, possibly because these approaches mainly encourage local robustness around training points. In such settings, our guarantee is less meaningful, and one would likely need local verification on each test example in order to guarantee robustness to all adversaries, a possibly costly procedure. Nevertheless, our guarantee can be made more useful in such settings as well by explicitly controlling the upper bound with spectral norm constraints.

## 3.2 Regularization of generative adversarial networks

Generative adversarial networks attempt to learn a *generator* neural network $G_\phi : \mathcal{Z} \to \mathcal{X}$, so that the distribution of $G_\phi(z)$ with $z \sim D_z$ a noise vector resembles a data distribution $D_x$. Various recent approaches have relied on regularization strategies on a *discriminator* network in order to improve the stability of GAN training and the quality of the produced samples. Some of these resemble our approaches from Section 2 such as gradient penalties (Gulrajani et al., 2017; Roth et al., 2017) and spectral norm regularization (Miyato et al., 2018a).

While these regularization methods had different motivations originally, we suggest that viewing them as approximations of an RKHS norm constraint provides some insight into their effectiveness in practice; specifically, we provide an interpretation as optimizing a kernel two-sample test such as MMD with the convolutional kernel introduced in Section 2:

$$\min_{\phi} \sup_{f \in \mathcal{H}: \|f\|_{\mathcal{H}} \leq 1} \mathbb{E}_{x \sim D_x}[f(x)] - \mathbb{E}_{z \sim D_z}[f(G_\phi(z))]. \tag{12}$$

In contrast to the Wasserstein GAN interpretation of a similar objective where all 1-Lipschitz functions are considered (Arjovsky et al., 2017), the MMD interpretation yields a parametric statistical rate $O(n^{-1/2})$ when learning from an empirical distribution with $n$ samples, which is significantly better than the $O(n^{-1/d})$ rate of the Wasserstein-1 distance for high-dimensional data such as images (Sriperumbudur et al., 2012). While the MMD approach has been used for training generative models, it generally relies on a generic kernel function, such as a Gaussian kernel, that appears explicitly in the objective (Dziugaite et al., 2015; Li et al., 2017; Bińkowski et al., 2018). Although using a learned feature extractor can improve this, the Gaussian kernel might be a poor choice when dealing with natural signals such as images, while the hierarchical kernel of Bietti & Mairal (2018) is better suited for this type of data, by providing useful invariance and stability properties. Leveraging the variational form of the MMD (12) with this kernel suggests using convolutional networks as the discriminator $f$, with constraints on the spectral norms in order to ensure $\|f\|_{\mathcal{H}} \leq C$ for some $C$, as done by Miyato et al. (2018a) through normalization.

Table 1: Regularization on CIFAR10 with 5 000 or 1 000 examples for VGG-11 and ResNet-18. Each entry shows the test accuracy with/without data augmentation when all hyper-parameters are optimized on a validation set.

| Method | 5k VGG-11 | 5k ResNet-18 | 1k VGG-11 | 1k ResNet-18 |
|---|---|---|---|---|
| No weight decay | 71.13/57.88 | 68.14/53.01 | 50.68/42.82 | 42.19/38.18 |
| Weight decay | 73.13/57.71 | 71.59/54.71 | 51.16/44.11 | 43.93/37.84 |
| SN penalty (PI) | 74.23/62.26 | 73.25/53.19 | 53.38/45.29 | 46.93/38.67 |
| SN penalty (SVD) | 74.7/60.87 | 73.49/55.46 | 52.32/44.52 | 47.03/40.04 |
| SN projection | 74.58/**63.36** | 75.87/55.31 | 53.52/46.0 | 45.11/38.56 |
| $\|f\|_M^2$ penalty | 72.95/60.16 | 70.61/55.47 | 51.71/45.0 | 45.23/**44.44** |
| $\|\nabla f\|^2$ penalty | 73.45/60.55 | 73.01/**57.45** | 54.32/**46.48** | **48.86**/44.0 |
| VAT | 72.45/59.16 | 73.07/55.75 | 52.06/43.21 | 46.6/42.46 |
| PGD-$\ell_2$ | 72.7/59.37 | 72.72/54.99 | 51.52/43.74 | 46.83/39.3 |
| PGD-$\ell_\infty$ | 73.29/58.04 | 72.81/54.13 | 51.92/43.74 | 46.27/40.97 |
| grad-$\ell_1$ | **75.13**/58.92 | 71.11/54.46 | 53.86/43.65 | 48.49/41.01 |
| grad-$\ell_2$ | **75.13**/58.63 | 73.83/55.39 | **54.79**/42.73 | 47.56/42.06 |
| PGD-$\ell_2$ + SN projection | **75.02**/61.42 | **76.06**/56.99 | 53.74/45.93 | 46.96/40.19 |
| grad-$\ell_2$ + SN projection | 74.67/63.12 | **76.17**/55.53 | **55.98**/**46.68** | **48.94**/43.26 |
| $\|f\|_M^2$ + SN projection | 74.08/**63.52** | 75.23/**58.8** | 54.03/**46.65** | 47.86/42.88 |
| $\|\nabla f\|^2$ + SN projection | 72.33/63.25 | 75.89/56.86 | **54.8**/**46.73** | **48.63**/**44.57** |

## 4 EXPERIMENTS

We tested the regularization strategies presented in Section 2 in the context of improving generalization on small datasets and training adversarially robust models. Our goal is to use common convolutional architectures used for large datasets and improve their performance in different settings through regularization.

For the adversarial training strategies, the inner maximization problems are solved with 5 steps of projected gradient ascent (Madry et al., 2018) with a randomly chosen starting point. In the case of the multi-class lower bound penalties $\|f\|_M^2$ and $\|\nabla f\|^2$ (see Section 2.4), we also maximize over examples in the mini-batch, only considering the maximal element when computing gradients w.r.t. parameters. For the robust optimization problem (7), we consider the PGD approach for $\ell_2$ and $\ell_\infty$ perturbations (Madry et al., 2018), as well as the corresponding $\ell_2$ (squared) and $\ell_1$ gradient norm penalties. For the upper bound approaches with spectral norms (SNs), we consider the SN projection strategy with decaying $\tau$, as well as the SN penalty (9), either using power iteration (PI) or a full SVD for computing gradients.

### 4.1 IMPROVING GENERALIZATION ON SMALL DATASETS

In this setting, we use 1 000 and 5 000 examples of the CIFAR10 dataset, with or without data augmentation. We consider a VGG network (Simonyan & Zisserman, 2014) with 11 layers, as well as a residual network (He et al., 2016) with 18 layers, which achieve 91% and 93% test accuracy respectively when trained on the full training set with data augmentation. We do not use any batch normalization layers in order to prevent any interaction with spectral norms. Each regularization strategy derived in Section 2 is trained for 500 epochs using SGD with momentum and batch size 128, halving the step-size every 40 epochs from a fixed initial step-size (0.05 for VGG-11, 0.1 for ResNet-18), a strategy we found to work relatively well for all methods. In order to study the potential effectiveness of each method, we assume that a fairly large validation set is available to select hyper-parameters; thus, we keep 10 000 annotated examples for this purpose.

Table 1 shows the test accuracies we obtain for upper and lower bound approaches, as well as combined approaches and different geometries. We also include virtual adversarial training (VAT, Miyato et al., 2018b). Among upper bound strategies, the constrained approach often works best, perhaps thanks to a more explicit control of the SNs compared to the

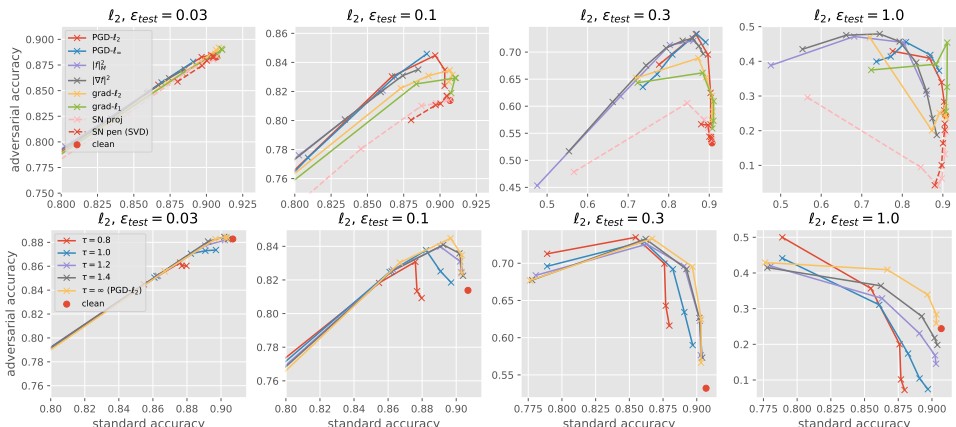

Figure 1: Robustness trade-off curves of different regularization methods for VGG11 on CIFAR10. Each plot shows test accuracy vs adversarial test accuracy for $\ell_2$-bounded, 40-step PGD adversaries with a fixed $\epsilon_{\text{test}}$. The bottom plots consider PGD-$\ell_2$ + SN projection, with different values of the constraint radius $\tau$. Different points on a curve correspond to training with different regularization strengths, with the leftmost points corresponding to the strongest regularization.

penalized approach. The SN penalty can work well nevertheless, and provides computational benefits when using the PI variant. For lower bound approaches, we found our $\|f\|_M^2$ and $\|\nabla f\|^2$ penalties to often work best, particularly without data augmentation, while the robust optimization gradient $\ell_1$ and $\ell_2$ norm penalties can be preferable with data augmentation. We note that the best regularization parameters are often quite small on such datasets, making the gradient penalties good approximations of the robust objective (7) used by PGD. Because gradient penalties have closed form gradients while PGD only obtains them by solving (7) approximately, they may work better in this setting thanks to optimization benefits. The adaptive nature of the regularization through robust optimization may also be beneficial on these datasets which are often easily separable with CNNs, however the explicit penalization by $\|f\|_M^2$ or $\|\nabla f\|^2$ seems to be more helpful in the case of 1 000 examples with no data augmentation, which is plausibly the hardest setting in our experiments. Finally, we can see that the combined approaches of lower bounds (either robust optimization, or separate penalties) together with SN constraints often yield the best results. Indeed, lower bound approaches alone do not necessarily control the upper bounds (and this is particularly true for PGD, as discussed in Section 4.2), which might explain why the additional constraints on SNs are helpful.

## 4.2 TRAINING ADVERSARIALLY ROBUST MODELS

We consider the same VGG architecture as in Section 4.1, trained on CIFAR10 with data augmentation, with different regularization strategies. Each method is trained for 300 epochs using SGD with momentum and batch size 128, dividing the step-size in half every 30 epochs. This strategy was successful in reaching convergence for all methods.

Figure 1 shows the test accuracy of the different methods in the presence of $\ell_2$-bounded adversaries, plotted against standard accuracy. We can see that the robust optimization approaches tend to work better in high-accuracy regimes, perhaps because the local stability and data fit that they encourage are sufficient on this dataset, while the $\|f\|_M^2$ penalty can be useful in some regimes where robustness to large perturbations is needed. We find that upper bound approaches alone do not provide robust models, but Figure 1(bottom) shows that combining the SN projection approach with a lower bound strategy (in this case PGD-$\ell_2$) helps improve robustness, perhaps thanks to a better control of margins and stability. The plots also confirm that gradient penalties are preferable for small regularization strengths (they achieve higher accuracy while improving robustness for small $\epsilon_{test}$), possibly due to

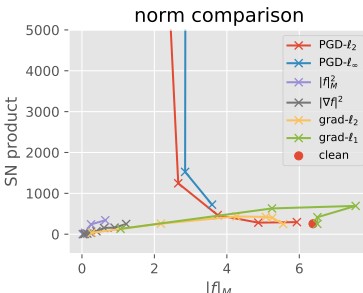 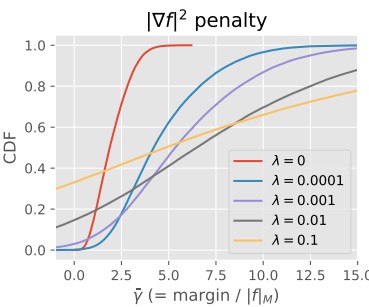

Figure 2: (left) Comparison of lower and upper bound quantities ($\|f\|_M$ vs the product of spectral norms). (right) CDF plot of normalized empirical margins for the $\|\nabla f\|^2$ penalty with different regularization strengths, normalized by $\|f\|_M$. We consider 1000 fixed training examples when computing $\|f\|_M$.

better optimization, while for stronger regularization, the gradient approximation no longer holds and the adversarial training approaches such as PGD are preferred.

These results suggest that a robust optimization approach such as PGD can work very well on the CIFAR10 dataset, perhaps because the training data is easily separable with a large margin and the method adapts to this "easiness". However, this raises the question of whether the approach is actually controlling $\|f\|_{\mathcal{H}}$, given that it only attempts to minimize a lower bound, as discussed in Section 2.2. Figure 2(left) casts some doubt on this, showing that for PGD in contrast to other methods, the product of spectral norms (representative of an upper bound on $\|f\|_{\mathcal{H}}$) increases when the lower bound $\|f\|_M$ decreases. This suggests that a network learned with PGD with large $\epsilon$ may have large RKHS norm, possibly because the approach tries to separate $\epsilon$-balls around the training examples, which may require a more complex model than simply separating the training examples (see also Madry et al., 2018). This large discrepancy between upper and lower bounds highlights the fact that such models may only be stable locally near training data, though this happens to be enough for robustness on many test examples on CIFAR10. Moreover, guaranteeing robustness at test time against all adversaries would likely require expensive verification procedures, given that the global guarantee given by the product of spectral norms is weak in this case.

In constrast, for other methods, and in particular the lower bound penalties $\|f\|_M^2$ and $\|\nabla f\|^2$, the upper and lower bounds appear more tightly controlled, suggesting a more appropriate control of the RKHS norm. This makes our guarantees on adversarial generalization more meaningful, and thus we may look at the empirical distributions of normalized margins $\bar{\gamma}$ obtained using $\|f\|_M$ for normalization, shown in Figure 2(right). The curves suggest that for small $\bar{\gamma}$, smaller values of $\lambda$ are preferred, while stronger regularization helps when $\bar{\gamma}$ increases, yielding lower test error when an adversary is present according to our bounds in Section 3.1. This qualitative behavior is indeed observed in the results of Figure 1 on test data for the $\|\nabla f\|^2$ penalty approach.

## 5 DISCUSSION

Making generic machine learning techniques more data efficient is crucial to reduce the costs related to annotation. While other approaches may also be important to solve this grand challenge, such as incorporating more prior knowledge in the architecture (*e.g.*, Oyallon et al., 2017), semi-supervised learning (Chapelle et al., 2006) or meta-learning (when multiple tasks or datasets are available, see, *e.g.*, Thrun, 1998), basic regularization principles will be needed and those are crucially missing today. Such principles are also essential for obtaining robust models in applications where security is a concern, such as self driving cars. Our paper presents various algorithmic strategies for regularization on generic deep convolutional networks, by leveraging the structure of an appropriate RKHS, leading to many existing approaches to regularization, as well as new ones, in addition to providing theoretical guarantees and insights on different methods.

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

## A    DETAILS ON OPTIMIZATION WITH SPECTRAL NORMS

This section details our optimization approach presented in Section 2.3 for learning with spectral norm constraints. In particular, we rely on a *continuation* approach, decreasing the size of the ball constraints during training, towards a final value $\tau$. The method is presented in Algorithm 1. We use an exponentially decreasing schedule for $\tau$, and found using 2 epochs for $\kappa$ to work well in practice. In the context of convolutional networks, we simply consider the SVD of a reshaped filter matrix, but we note that alternative approaches based on the singular values of the full convolutional operation may also be used (Sedghi et al., 2018).

## B    EXTENSIONS TO NON-EUCLIDIAN GEOMETRIES

The kernel approach from previous sections is well-suited for input spaces $\mathcal{X}$ equipped with the Euclidian distance, thanks to the non-expansiveness property (2) of the kernel mapping. In the case of linear models, this kernel approach corresponds to using $\ell_2$-regularization

---

**Algorithm 1** Stochastic projected gradient with continuation

---

Input: $\tau$, $\kappa$, step-sizes $\eta_t$
**for** $t = 1, \ldots$ **do**
    Sample mini-batch and compute gradients of the loss w.r.t. each $W^l$, denoted $G_t^l$
    $\tau_t = \tau(1 + \exp\left(\frac{-t}{\kappa}\right))$
    **for** $l = 1, \ldots, L$ **do**
        $\tilde{W}_t^l := W_t^l - \eta_t G_t^l$
        Compute SVD: $\tilde{W}_t^l = U\mathrm{diag}(\sigma)V^T$
        $\widehat{\sigma} := \mathrm{proj}_{\|.\|_\infty \leq \tau_t}(\sigma)$
        $W_{t+1}^l := U\mathrm{diag}(\widehat{\sigma})V^T$
    **end for**
**end for**

---

by taking a linear kernel. However, other forms of regularization and geometries can often be useful, for example to encourage sparsity with an $\ell_1$ regularizer. Such a regularization approach presents tight links with robustness to $\ell_\infty$ perturbations on input data, thanks to the duality relation $\|w\|_1 = \sup_{\|u\|_\infty} \langle w, u \rangle$ (see Xu et al., 2009a).

In the context of deep networks, we can leverage such insights to obtain new regularizers, expressed in the same variational form as the lower bounds in Section 2.2, but with different geometries on $\mathcal{X}$. For $\ell_\infty$ perturbations, we obtain

$$\sup_{x,y\in\mathcal{X}} \frac{f(x) - f(y)}{\|x - y\|_\infty} \quad \geq \quad \sup_{x\in\mathcal{X}} \|\nabla f(x)\|_1. \tag{13}$$

The Lipschitz regularizer (l.h.s.) can also be taken in an adversarial perturbation form, with $\ell_\infty$-bounded perturbations $\|\delta\|_\infty \leq \epsilon$. When considering the corresponding robust optimization problem

$$\min_\theta \frac{1}{n} \sum_{i=1}^n \sup_{\|\delta\|_\infty \leq \epsilon} \ell(y_i, f_\theta(x_i + \delta)), \tag{14}$$

we may consider the PGD approach of Madry et al. (2018), or the associated gradient penalty approach with the $\ell_1$ norm, which is a good approximation when $\epsilon$ is small (Lyu et al., 2015; Simon-Gabriel et al., 2018).

As most visible in the gradient $\ell_1$-norm in (13), these penalties encourage some sparsity in the gradients of $f$, which is a reasonable prior for regularization on images, for instance, where we might only want predictions to change based on few salient pixel regions. This can lead to gains in interpretability, as observed by Tsipras et al. (2018).

We note that in the case of linear models, our robust margin bound of Section 3.1 can be adapted to $\ell_\infty$-perturbations, by leveraging Rademacher complexity bounds for $\ell_1$-constrained models (Kakade et al., 2009). Obtaining similar bounds for neural networks would be interesting but goes beyond the scope of this paper.

**Experiments with $\ell_\infty$ adversaries.** Figure 3 shows similar curves to Figure 1 from Section 4.2, but where the attacker is constrained in $\ell_\infty$ norm instead of $\ell_2$ norm. We can see that using the right metric in PGD indeed helps against an $\ell_\infty$ adversary, nevertheless controlling global stability through the RKHS norm as in the $\|f\|_M^2$ and $\|\nabla f\|^2$ penalties can still provide some robustness against such adversaries, even with large $\epsilon_t est$. For gradient penalties, we find that the different geometries behave quite similarly, which may suggest that more appropriate optimization algorithms than SGD could be needed to better accommodate the non-smooth case of $\ell_1/\ell_\infty$, or perhaps that both algorithms are actually controlling the same notion of complexity on this dataset.

## C DETAILS ON GENERALIZATION GUARANTEES

This section presents the proof of Proposition 1, which relies on standard tools from statistical learning theory (*e.g.*, Boucheron et al., 2005).

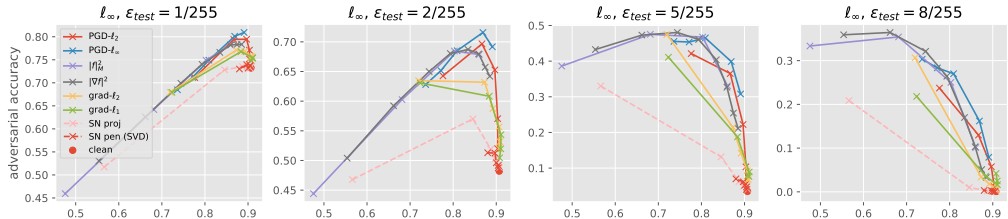

Figure 3: $\ell_\infty$ robustness trade-off curves of different regularization methods for VGG11 on CIFAR10. Each plot shows test accuracy vs adversarial test accuracy for $\ell_\infty$ bounded 40-step PGD adversaries with a fixed $\epsilon_{\text{test}}$. Different points on a curve correspond to training with different regularization strengths, with the leftmost points corresponding to the strongest regularization.

### C.1 Proof of Proposition 1

*Proof.* Assume for now that $\gamma$ is fixed in advance, and let $\mathcal{F}_\lambda := \{f \in \mathcal{H} : \|f\|_{\mathcal{H}} \le \lambda\}$. Note that for all $f \in \mathcal{F}_\lambda$ we have

$$\text{err}_{\mathcal{D}}(f, \epsilon) = P(\exists \|\delta\| \le \epsilon : yf(x + \delta) < 0) \le P(yf(x) < \lambda\epsilon) =: L^{\lambda\epsilon}(f),$$

since $\|f\|_{\mathcal{H}} \le \lambda$ is an upper bound on the Lipschitz constant of $f$. Consider the function

$$\phi(x) = \begin{cases} 0, & \text{if } x \le -\gamma - \lambda\epsilon \\ 1, & \text{if } x \ge -\lambda\epsilon \\ 1 + (x + \lambda\epsilon)/\gamma, & \text{otherwise.} \end{cases}$$

Defining $A(f) = \mathbb{E}\,\phi(-yf(x)) \ge L^{\lambda\epsilon}(f)$ and $A_n(f) = \frac{1}{n}\sum_{i=1}^{n}\phi(-y_i f(x_i)) \le L_n^{\lambda\epsilon+\gamma}(f)$, and noting that $\phi$ is upper bounded by 1 and $1/\gamma$ Lipschitz, we can apply similar arguments to (Boucheron et al., 2005, Theorem 4.1) to obtain, with probability $1 - \delta$,

$$L^{\lambda}\epsilon(f) \le L_n^{\lambda\epsilon+\gamma}(f) + O\left(\frac{1}{\gamma}R_n(\mathcal{F}_\lambda) + \sqrt{\frac{\log 1/\delta}{n}}\right),$$

where $R_n(\mathcal{F}_\lambda)$ denotes the empirical Rademacher complexity of $\mathcal{F}_\lambda$ on the dataset $\{(x_i, y_i)\}_{i=1,\dots,n}$. Standard upper bounds on empirical Rademacher complexity of kernel classes with bounded RKHS norm yield the following bound

$$\text{err}_{\mathcal{D}}(f, \epsilon) \le L_n^{\lambda\epsilon+\gamma}(f) + O\left(\frac{\lambda}{\gamma\sqrt{n}}\sqrt{\frac{1}{n}\sum_{i=1}^{n}K(x_i, x_i)} + \sqrt{\frac{\log 1/\delta}{n}}\right).$$

Note that the bound is still valid with $\gamma' \ge \gamma$ instead of $\gamma$ in the first term of the r.h.s., since $L_n^{\gamma}(f)$ is non-decreasing as a function of $\gamma$.

In order to establish the final bound, we instantiate the previous bound for values $\lambda_i = 2^i$ and $\gamma_j = 2^{-j}$. Defining $\delta_{i,j} = \frac{\delta}{(1+4i^2)\cdot(1+4j^2)}$, we have that w.p. $1 - \delta_{i,j}$, for all $f \in \mathcal{F}_{\lambda_i}$ and all $\gamma \ge \gamma_j$,

$$\text{err}_{\mathcal{D}}(f, \epsilon) \le L_n^{\lambda_i\epsilon+\gamma}(f) + O\left(\frac{\lambda_i}{\gamma_j\sqrt{n}}\sqrt{\frac{1}{n}\sum_{i=1}^{n}K(x_i, x_i)} + \sqrt{\frac{\log 1/\delta_{i,j}}{n}}\right). \tag{15}$$

By a union bound, this event holds jointly for all integers $i, j$ w.p. greater than $1 - \delta$, since $\sum_{i,j}\delta_{i,j} \le \delta$. Now consider an arbitrary $f \in \mathcal{H}$ and $\gamma > 0$ and let $i = \lceil \log_2\|f\|_{\mathcal{H}}\rceil$ and $j = \lceil \log_2(1/\gamma)\rceil$. We have

$$\lambda_i \le 2\|f\|_{\mathcal{H}}$$
$$\frac{1}{\gamma_j} \le \frac{2}{\gamma}$$
$$\log(1/\delta_{i,j}) \le \log(C(\|f\|_{\mathcal{H}}, \gamma)/\delta).$$

Applying this to the bound in (15) yields the desired result.

$\square$

