# OpenReview forum: "On Regularization and Robustness of Deep Neural Networks"
_ICLR.cc/2019/Conference_

### Official Review · AnonReviewer2 · 2018-10-23

**Rating:** 6
**Confidence:** 2

**Review:**

In this paper, the authors consider CNN models from the lens of kernel methods. They build upon past work that showed that such models can be seen to lie in appropriate RKHS, and derive upper and lower bounds for the kernel norm. These bounds can be used as regularizers that help train more robust neural networks, especially in the context of euclidean perturbations of the inputs, and training GANs. They show that the bounds can also be used to recover existing special cases such as spectral norm penalizations and gradient regularization. They derive generalization bounds from the point of view of adversarial learning, and report experiments to buttress their claims.

Overall, the paper is a little confusing. A lot of the times, the result seem to be a derivative of the work by Bietti and Mairal, and looks like the main results in this paper are intertwined with stuff B+M already showed in their paper. It's hard to ascertain what exactly the contributions are, and how they might not be a straightforward consequence of prior work (for example, combining results from Bietti and Mairal; and generalization bounds for linear models). It might be nice to carefully delineate the authors' work from the former, and present their contributions.

Page 4: Other Connections with Lower bounds: The first line " "we may also consider ... ". This line is vague. How will you ensure the amount of deformation is such that the set \bar{U} is contained in U ?

Page 4 last paragraph: "One advantage ... complex architectures in practice" : True, but the tightness of the bounds *do* depend on "f" (specifically the RKHS norm). It needs to be ascertained when equality holds in the bounds you propose, so that we know how tight they are. What if the bounds are too loose to be practical?

eqn (8): use something else to denote the function 'U'. You used 'U' before to denote the set.

eqn (12): does \tilde{O} hide polylog factors? please clarify.

---

> ### Author Response · Authors · 2018-11-08
> **Response**
>
> We thank the reviewer for his comments. We discuss the novelty aspects in our general response ( https://openreview.net/forum?id=HkMlGnC9KQ&noteId=S1eid00WaQ ) and will be happy to clarify this in the paper. Further comments are addressed below.
>
> ** controlling the amount of deformations
>
> The stability bounds of B+M provide upper bounds on ||Phi(x') - Phi(x)|| (where x' is a deformation of x) based on quantities related to the corresponding diffeomorphism, i.e. the maximum norm and the maximum jacobian norm. For simple classes of deformations these can be computed precisely in terms of the parameters of the deformation, e.g. for translations, rotations, scaling or simple parametric warps. When bounding these away from zero by a certain constant, ||Phi(x') - Phi(x)|| is then included in a centered ball of the RKHS with a radius growing with this constant. This constant then acts as a regularization parameter, just like the size of additive perturbations in the case of adversarial perturbations, and can be tuned by cross-validation.
>
> ** tightness of the lower bounds
>
> This is something that we verify empirically in our experiments at the end of training by checking the values of spectral norms as a proxy of the upper bound, and looking at the gap with the lower bound. In particular, when using the ||f||_M penalty, lower and upper bounds seem to be controlled together in our experiments (Figure 2), making the bound useful, in contrast to PGD, for which spectral norms grow uncontrolled when the lower bound decreases. We will further clarify this in the paper.
>
> eqn (8), (12): thanks for pointing these out, we will fix this in the paper.

---

### Official Review · AnonReviewer3 · 2018-11-04
**Interesting ideas, but not enough of an independent contribution**

**Rating:** 4
**Confidence:** 4

**Review:**

This paper looks at adversarial examples from the context of RKHS norms for neural networks.  The work builds conceptually on the work of Bietti and Mairal (2018), who investigate approximate RKHS norms for neural networks (including computation via a specialized convolutional kernel), and Xu et al., (2009) which looks at robustness properties of kernel classifiers.  The authors discuss how the RKHS norm of neural network functions provide robustness guarantees for the resulting classifier, both in terms of a straightforward robustness property for a given example, as well as in terms of generalization guarantees about robustness.

Overall, I think there are some interesting ideas in this work, but ultimately not enough to make a compelling independent paper.  The core issue here is that the RKHS properties are used only in a very minimal manner to actually provide much analysis or insight into the robustness properties of the network.  For example, the upper bound in (8) seems to be central here to illustrating how a bound on the RKHS norm can be upper bounded as a function of the operator l2 norm of the inner weight matrices (though the actual form of the bound isn't mentioned), and the latter term could thus provide a certified bound on the robustness loss of a classifier.  However, there are two big issues here: 1) it's trivial to directly bound the l2 robustness of a classifier by the product of the weight spectral norms and 2) the actual regularization term the authors proposed to use (the sum of spectral norms) is notably _not_ an upper bound on either the robust loss or the RKHS norm; naturally, this penalty along with the constrained version will still provide some degree of control over the actual robustness, but the authors don't connect this to any real bound.  I also think the authors aren't properly acknowledging just how similar this is to past work: the Parseval networks paper (Cisse et al., 2017), for instance, presents a lot of similar discussion of how to bound generalization error based based upon terms involving operator norms of the matrices, and the actual spectral normalization penalty that the authors advocate for has been studied by Miyato et al. (2018).  To be clear, both of these past works (and several similar ones) are of course cited by the current paper, but from a practical standpoint it's just not clear to me what the takeaways should be here above and beyond this past work, other than the fact that these quantities _also_ bound the relevant RKHS norms.  Likewise the generalization bound in the paper is a fairly straightforward application of existing bounds given the mechanics of the RKHS norm defined by previous work.

To be clear, I think the RKHS perspective that the authors advocate for here is actually quite interesting.  I wasn't particularly familiar with the Bietti and Mairal (2018) work, and going through it in some detail for reviewing this paper, I think it's an important directly for analysis of deep networks, including from a perspective of robustness.  But the results here seem more like a brief follow-on note to the past work, not a complete set of results in and of themselves.  Indeed, because the robustness perspective here can largely be derived completely independently of the RKHS framework, and because the resulting training procedures seem to be essentially identical to previously-proposed approaches, the mere contribution of connecting these works to the RKHS norm doesn't seem independently to be enough of a contribution in my mind.

One final, though more minor, point: It's worth pointing out that (globally) bounding the Lipschitz constant seems top stringent a condition for most networks, and most papers on certifiable robustness seem to instead focus on some kind of local Lipschitz bound around the training or test examples.  Thus, it's debatable whether even the lower bound on the RKHS norm is really reasonable to consider for the purposes of adversarial robustness.

---

> ### Author Response · Authors · 2018-11-08
> **Response**
>
> We thank the reviewer for his comments. Our general response ( https://openreview.net/forum?id=HkMlGnC9KQ&noteId=S1eid00WaQ ) details the aspects related to novelty. Further comments are addressed below.
>
> ** comparison with Parseval networks + works of Miyato et al.
>
> We agree that a better comparison with the Parseval network paper would be useful. Regarding generalization, the Parseval networks paper seems to only discuss standard generalization performance based on robustness, not *adversarial* generalization (that is, test error in the presence of an adversary), as considered in our work. Also, our bound seems significantly better: whereas the bound in Cissé et al. (2017) has an exponential dependence on the dimension due to covering number of the input space (this is a weakness of the generalization bounds from Xu and Mannor (2012), which do not leverage statistical properties of the function class being used), our margin bound has no dependence on the dimension, or only a logarithmic dependence if we use the Rademacher analysis of Bartlett et al. (2017) instead of our kernel framework.
> Regarding the improper acknowledgement of Miyato's work, we are a bit surprised by the reviewer's comment: we cite the work of Miyato almost each time we mention spectral regularization and the acknowledgment seems clear to us throughout the paper. However, if the reviewer finds any ambiguous claim in our paper that we would have missed, we would be happy to clarify it.
>
>  ** role of the specific regularizer
>
> The reviewer points out that some of the regularization functions we consider such as the spectral norm penalties, are not based on the precise upper bound we derive. Whereas optimizing a product of spectral norms is impractical, which naturally leads to other variants (sums of spectral norms or constraints), we would like to emphasize that such variants are empirically effective in the sense that the quantities obtained at the end of training---such as the spectral norms, (local or global) Lipschitz constants, and the margins of each datapoint---are controlled. These quantities are also what governs our generalization guarantees. Besides, we also note that many deep architectures with ReLUs (particularly VGG-like, if we ignore bias terms) are homogeneous in the weight matrices, making the relative norms at each layer not crucial (multiplying one layer by a scalar and dividing another by the same scalar leads to an equivalent model). In particular, this justifies using the same value for the spectral norm constraint of each layer.
>
>  ** Usefulness of the RKHS framework
>
> The RKHS framework was quite beneficial in our work because it displays several properties at once:
>  (1) clear understanding of regularization and generalization through margin bounds
>  (2) makes a clear link between stability/robustness and regularization/generalization by using the RKHS norm and properties of the kernel mapping
>  (3) yields practical regularization algorithms through upper and lower bounds
>
> Looking at alternatives, if we consider the product of spectral norms instead of the RKHS norm, then we may have (1) using results of Bartlett et al.(2017) and partly (2) since we can upper bound the Lipschitz constant, however algorithms based on lower bounds are crucially missing, and our experiments suggest that these algorithms are often important for good performance, both for regularization on small datasets and for robustness.
> If instead we consider the robust optimization approach, we obtain variants of good algorithms (3) such as PGD or gradient penalties, and perhaps some connections to regularization following Xu et al. (2009), however it is difficult to obtain useful generalization guarantees without defining a precise quantity of model complexity. Additionally, such approaches may favor local over global robustness, particularly with powerful function approximators such as neural networks, which may be undesirable when one wants global guarantees.
>
> ** bounding l2 robustness with product of spectral norms
>
> It is indeed easy to upper bound l2 robustness using the product of spectral norms. However, such a robustness guarantee is only useful if this quantity is appropriately controlled during training. In particular, for methods like PGD, we find that such a quantity is poorly controlled on Cifar10, and would thus only provide very weak guarantees.
>
> ** on global vs local Lipschitz constants
>
> We agree that in some cases local robustness is enough in practice, however this may come at the cost of having weak guarantees on adversarial generalization, and may require expensive verification procedures locally around each test example for guaranteed robustness, as mentioned in our general response.
>
> We will happily clarify some of these points in an updated version of the paper.

---

### Official Review · AnonReviewer1 · 2018-11-05
**Well written with interesting findings, but limited novelty**

**Rating:** 5
**Confidence:** 3

**Review:**

Regularizing RKHS norm is a classic way to prevent overfitting. The authors
note the connections between RKHS norm and several common regularization and
robustness enhancement techniques, including gradient penalty, robust
optimization via PGD and spectral norm normalization. They can be seen as upper
or lower bounds of the RKHS norm.

There are some interesting findings in the experiments. For example, for
improving generalization, using the gradient penalty based method seems to work
best.  For improving robustness, adversarial training with PGD has the best
results (which matches the conclusions by Madry et al.); but as shown in Figure
2, because adversarial training only decreases a lower bound of RKHS norm, it
does not necessarily decrease the upper bound (the product of spectral norms).
This can be shown as a weakness of adversarial training if the authors explore
further and deeper in this direction.

Overall, this paper has many interesting results, but its contribution is
limited because:

1. The regularization techniques in reproducing kernel Hilbert space (RKHS) has
been well studied by previous literature. This paper simply applies these
results to deep neural networks, by treating the neural network as a big
black-box function f(x).  Many of the results have been already presented in
previous works like Bietti & Mairal (2018).

2. In experiments, the authors explored many existing methods on improving
generalization and robustness. However all these methods are known and not new.
Ideally, the authors can go further and propose a new regularization method
based on the connection between neural networks and RKHS, and conduct
experiments to show its effectiveness.

The paper is overall well written, and the introductions to RKHS and each
regularization techniques are very clear. The provided experiments also include
some interesting findings. My major concern is the lack of novel contributions
in this paper.

---

> ### Author Response · Authors · 2018-11-08
> **Response**
>
> We thank the reviewer for his comments. We address the comments about novelty in our general response ( https://openreview.net/forum?id=HkMlGnC9KQ&noteId=S1eid00WaQ ), for instance concerning the relationship to previous work, and the regularization penalty ||f||_M we propose. More detailed comments are addressed below.
>
> ** weakness of adversarial training
>
> As noted in our general response, our ||f||_M regularization approach empirically yields models with a more useful certified generalization guarantee in the presence of adversaries on Cifar10, while PGD adversarial training would likely require local verification of robustness around each test example, and we are not aware of useful guarantees on adversarial generalization for such models. We agree that this aspect is not clear in the current submission, and we will improve it in the next version.
>
> ** relationship with traditional RKHS regularization
>
> There is indeed no question that kernel methods/RKHSs have been widely used for regularization of non-linear functions, for over 20 years now, however these methods typically rely on solving convex optimization problems using the kernel trick, or various kernel approximations (such as random Fourier features). Separately, defining RKHSs that contain neural networks has indeed been the study of previous work, such as Bietti and Mairal (2018) or Zhang et al. (2016; 2017), however these only study theoretical properties of the kernel mapping and the RKHS norm, or derive convex learning procedures to replace training neural networks. Our approach is quite different, in that we leverage these insights to obtain practical regularization strategies for generic neural networks.
>
> ** new regularization methods
>
> In addition to the ||f||_M lower bound penalty discussed in our general response, we note that combined approaches based on lower bound + upper bound methods are also novel to the best of our knowledge, and in particular we found combining robust optimization techniques with spectral norm constraints to be quite successful in many of the small data scenarios considered (see Table 1).
>
> We will happily clarify some of these points in an updated version of the paper.

---

> > ### Comment · AnonReviewer1 · 2018-11-27
> > **Thanks for the response!**
> >
> > Dear Paper1233 Authors,
> >
> > After reading the response carefully, I still feel like this paper is not ready to publish. Part of the reason is the organization of this paper does not highlight its main contributions, and also the paper lacks in-depth original contribution.
> >
> > I encourage the authors to continue their works in this direction and reorganize the work to emphasize the main contributions. Especially, the lower bound+upper bound methods for RKHS regularization can be further developed and extended to a good work on its own.
> >
> > Thanks,
> > Paper1233 AnonReviewer1

---

### Author Response · Authors · 2018-11-08
**General response to reviewers**

First, we would like to thank the reviewers for their useful remarks and suggestions. We provide general comments here that are relevant for all reviewers, with specific comments for each reviewer in individual replies.

Based on the reviewers' comments, we realize that our original submission focuses too much on establishing links between regularization with the RKHS norm and existing strategies, which we found particularly interesting, rather than highlighting the benefits of the newly obtained strategies. Additionally, some of our observations and insights were a bit preliminary in the submission, particularly regarding robustness and security applications, given that our original main motivation for this work was different, focusing on regularization in small data settings. We hope to better clarify the concerns about novelty here, and welcome further remarks by the reviewers. We will do our best to update the paper accordingly before the end of the discussion period.

** Novelty compared to Bietti and Mairal (2018)

Bietti and Mairal (2018) study theoretical properties of the RKHS only. Here, we provide *practical algorithms* for regularizing usual neural networks using the RKHS norm, which is a major step forward compared to this existing work.  We are also not aware of previous work that considers the RKHS norm for regularization of deep networks in practice. An interesting insight of our work is that this norm is quite large for standard networks trained with SGD, and that explicitly controlling it brings clear benefits.

** Novelty of the regularization strategies

The adversarial perturbation penalty ||f||_M that we introduce in this work is quite different from previous work: (i) it encourages stability of the entire prediction function by considering a separate penalty term in the optimization objective; (ii) it optimizes worst-case stability across the domain, in contrast to other approaches which only optimize this on average over training points. On Cifar10, our method seems most effective in controlling the RKHS norm compared to other methods, where we observe that both the lower bound and spectral norms are controlled together. In contrast, other lower bound methods seem less effective at controlling the upper bound, and this is particularly pronounced in the case of PGD in to our experiments. This is mentioned in the last paragraph of Section 4, and displayed in Figure 2(left).
The approach ||f||_M yields the best accuracy for regularization on some of the small dataset problems we consider, and can achieve the best performance in some regimes of the robustness/generalization trade-off. Additionally, it provides the most useful certified guarantee on adversarial generalization in our experiments (see next bullet point).

Another key difference between our approach and previous ones is that our penalties involve a global optimization problem across the space of inputs X rather than only an average over training samples. This is true both for ||f||_M vs. PGD and for our gradient penalty vs. existing strategies based on gradients. We admittedly did not yet investigate the importance of this local vs. global regularization effect (and we currently only optimize across examples in a mini-batch), which we plan to do in a longer version of the paper. This indeed paves the way to transductive and semi-supervised settings, which we plan to investigate as well.

** Certified guarantees for adversarial generalization, and novelty of our theoretical analysis

We also would like to point out that our original main motivation was to study regularization benefits, while some of our observations regarding robustness and security were somewhat preliminary at the time of submission. Yet, one important aspect of our work in the context of robustness is that controlling the RKHS norm can provide a model with *certified* guarantees on *adversarial* generalization (i.e. test accuracy in the presence of an adversary), as given by our margin bound analysis, although it depends on the RKHS norm which can only be approximated. We note that while margin bounds have been useful to establish (standard) generalization guarantees for neural networks, to our knowledge our work is the first to use similar arguments for bounding adversarial generalization.

Our experiments suggest that the most useful guarantees are obtained for models trained with our penalty ||f||_M, for which the upper and lower bounds are more tightly related than for other methods (see Figure 2). In contrast, while methods like PGD may give improved robustness empirically in some regimes, our experiments on CIFAR10 suggest that the obtained models have large spectral norms, yielding quite weak guarantees on adversarial generalization.
This suggests that the robustness of such models may be only local, so that one may need (possibly costly) verification procedures on each test example in order to guarantee robustness against all adversaries.

---

### Author Response · Authors · 2018-11-26
**Update after revision**

We have updated the paper with clarifications on the novelty of our RKHS perspective to regularization, and emphasized the benefits of controlling the RKHS norm, an aspect which was not clear in the original submission. We also included additional empirical results.

In particular, we hope to have clarified the following points:
 * empirically, we find that an appropriate control of the RKHS norm seems to be missing for existing methods based on robust optimization (which can give up global stability in favor of local robustness) or spectral norms (which reduce model complexity but can remain unstable locally)
 * the penalties |f|_M^2 and |\nabla f|^2 that we obtain from RKHS arguments seem to provide a better control of the RKHS norm in practice
 * combining lower and upper bound approaches can further help control this norm
 * empirically, these methods often yield the best generalization performance on small datasets, and additionally can provide the most useful guarantees on adversarially robust generalization

We hope this update clarifies the concerns of the reviewers, and we would like to sincerely thank all reviewers again for their useful comments and remarks, which helped us improve our paper.

---

### Meta-Review · Area_Chair1 · 2018-12-15
**interesting perspective but insufficient contribution**

**Confidence:** 5
**Recommendation:** Reject

**Metareview:**

Reviewers generally found the RKHS perspective interesting, but did not feel that the results in the work (many of which were already known or follow easily from known theory) are sufficient to form a complete paper. Authors are encouraged to read the detailed reviewer comments which contain a number of critiques and suggestions for improvement.